# Dynamics of Ultracold Bosons in Artificial Gauge Fields—Angular Momentum, Fragmentation, and the Variance of Entropy

**DOI:** 10.3390/e23040392

**Published:** 2021-03-25

**Authors:** Axel U. J. Lode, Sunayana Dutta, Camille Lévêque

**Affiliations:** 1Physikalisches Institut, Albert-Ludwigs-Universität Freiburg, Hermann-Herder-Straße 3, D-79104 Freiburg, Germany; 2Department of Mathematics, University of Haifa, Haifa 3498838, Israel; sunayanadutta59@gmail.com; 3Haifa Research Center for Theoretical Physics and Astrophysics, University of Haifa, Haifa 3498838, Israel; 4Vienna Center for Quantum Science and Technology, Atominstitut, TU Wien, Stadionallee 2, 1020 Vienna, Austria; camille.leveque@tuwien.ac.at; 5Wolfgang Pauli Institute c/o Faculty of Mathematics, University of Vienna, Oskar-Morgenstern Platz 1, 1090 Vienna, Austria

**Keywords:** Boson systems, ultracold gases, trapped gases, dynamic properties of condensates, collective and hydrodynamic excitations, superfluid flow, Bose–Einstein condensates, vortices, topological excitations

## Abstract

We consider the dynamics of two-dimensional interacting ultracold bosons triggered by suddenly switching on an artificial gauge field. The system is initialized in the ground state of a harmonic trapping potential. As a function of the strength of the applied artificial gauge field, we analyze the emergent dynamics by monitoring the angular momentum, the fragmentation as well as the entropy and variance of the entropy of absorption or single-shot images. We solve the underlying time-dependent many-boson Schrödinger equation using the multiconfigurational time-dependent Hartree method for indistinguishable particles (MCTDH-X). We find that the artificial gauge field implants angular momentum in the system. Fragmentation—multiple macroscopic eigenvalues of the reduced one-body density matrix—emerges in sync with the dynamics of angular momentum: the bosons in the many-body state develop non-trivial correlations. Fragmentation and angular momentum are experimentally difficult to assess; here, we demonstrate that they can be probed by statistically analyzing the variance of the image entropy of single-shot images that are the standard projective measurement of the state of ultracold atomic systems.

## 1. Introduction

Since the first realization of Bose-Einstein condensates in 1995 [1,2,3], ultracold atoms have become a standard probe for analog quantum simulations—due to the tunability and flexibility of these quantum states of matter, they can be manipulated to behave like other systems, for instance, condensed matter systems which are not as flexible or easy to observe. Popular examples include the realization of the quantum simulation of the superfluid-to-Mott-insulator transition [4,5], quantized conductance [6,7], the Dicke model [8,9], and magnetism realized via artificial gauge fields for ultracold atoms [10].

Such artificial gauge fields can make the neutral ultracold atoms behave as if they were charged particles experiencing a magnetic field and were investigated experimentally and theoretically with an external lattice potential [11,12,13] or without one [14,15,16].

In this paper, we investigate the physics of a two-dimensional system of harmonically trapped interacting ultracold bosons quenched with an artificial magnetic field (AMF) from a *many-body* point of view. The time-dependent Gross-Pitaevskii mean-field theory [17,18] is the most widespread tool to theoretically model many-body systems of ultracold bosonic atoms subject to an AMF. This approach recovers many of the physical phenomena observed, but neglects correlations by its construction using a mean-field ansatz; here, we go beyond mean-field and use the multiconfigurational time-dependent Hartree method for bosons (MCTDH-B) [19,20,21] to approximate the solution of the Schrödinger equation for ultracold atoms subject to an AMF. MCTDH-B is a method from the MCTDH family of methods [22,23,24,25,26,27] for indistinguishable particles (MCTDH-X) [22,28,29,30,31,32,33,34,35,36,37,38] that is able to self-consistently describe physics involving the presence and effects of quantum correlations. MCTDH-X was successfully applied to demonstrate the importance of variances of observables [39,40,41,42,43] and of single-shot images [9,44] to the correlations of particles in the many-body state. Using MCTDH-X, intriguing correlation effects beyond the commonly employed Bose-Hubbard description were found to be present in lattices [45,46,47,48,49,50] and cavity-generated lattices [51,52,53] and the breakdown of commonly used mean-field approaches has been demonstrated in tunneling dynamics [54,55,56] and in harmonic traps [57,58,59]. A key focus of the applications of MCTDH-B has been the emergence of fragmentation [60,61,62], where the reduced one-body density matrix has multiple significant eigenvalues, see, for instance, Refs. [54,63,64,65,66,67,68,69,70]. To obtain the results presented in this work, we used the MCTDH-X software hosted at http://ultracold.org (accessed on π-day, 14 March 2021), see References [38,50,71,72,73,74].

Our paper is structured as follows—in Section 2 we introduce the Hamiltonian and the MCTDH-X method we use, in Section 3 we discuss the observables that we are using in Section 4 to investigate the dynamics of ultracold atoms in an AMF; Section 5 summarizes our conclusions and provides an outlook.

## 2. Hamiltonian and Methods

We consider a system of bosonic particles with two-body interactions in two spatial dimensions. The state of the bosons is initialized in the ground state of a parabolic trap without an AMF present. Subsequently, the system is quenched by turning on suddenly an artificial gauge field corresponding to a homogeneous AMF perpendicular to the plane in which the bosons are trapped.

For the sake of clarity of presentation, we will omit the dependence of quantities on time *t* throughout this work, where it is obvious.

### 2.1. Setup

To setup the time-dependent many-body Schrödinger equation (TDSE), we use the Hamiltonian
(1)$$H=\int dx\;{\hat{\Psi }}^{†}\left(x\right)\left[T(x)+V(x)\right]\hat{\Psi }\left(x\right)\;+\frac{1}{2}\int dxdx^{\prime }\;{\hat{\Psi }}^{†}\left(x\right){\hat{\Psi }}^{†}\left(x^{\prime }\right)W\left(x,x^{\prime }\right)\Psi \left(x^{\prime }\right)\hat{\Psi }\left(x\right).$$Here, we work in atomic units (ℏ=m=1), the potential V(x) [with x=(x,y)] is chosen to be harmonic, $V\left(x\right)=\frac{1}{2}x^{2}$, and we consider contact interactions $W\left(x,x^{\prime }\right)=g_{0}\delta \left(x-x^{\prime }\right)$. Formally, one cannot use a contact interaction in two spatial dimensions with a complete basis set since the outcome would be that of the noninteracting bosons [75,76]. In simple terms, this is because the integral measure of the support of the Dirac-δ is zero for two and more spatial dimensions. For a proof, the interested reader is deferred to Ref. [76] and for an example for finding non-zero ranged Gaussian interaction potentials with similar physical behavior for ultracold bosons, see Ref. [75].

In the present work, we employ a finite truncation of the many-body basis (chiefly M=4 orbitals) and aim to demonstrate that beyond-mean-field phenomena do emerge. The kinetic energy is augmented with an artificial gauge field A(x;t):(2)$$\hat{T}\left(x\right)=\frac{1}{2}{\left(-i\nabla_{x}-gA\left(x;t\right)\right)}^{2}.$$For simplicity, we consider the case of unit charge g=1 and a homogeneous magnetic field B in *z*-direction of strength B(x;t):(3)$$B\left(x;t\right)=B\left(t\right){\hat{e}}_{z}.$$Here, ${\hat{e}}_{z}$ denotes the unit vector in *z*-direction. In the following, we work in Landau gauge,
(4)$$A\left(x;t\right)=B\left(t\right){\hat{e}}_{x},$$
and consider a quench scenario in the following, that is,
(5)B(t)=BΘ(t).Here, Θ(t) denotes the Heaviside step function, that is, the magnetic field is suddenly turned on at t>0 after the system has been initialized, see Figure 1 for a sketch.

In our investigation, we analyze the many-body dynamics by monitoring observables as a function of the effective magnetic field strength *B* at t>0 after the quench.

### 2.2. Method

To solve the TDSE,
(6)$$H|\Psi \rangle =i\partial_{t}|\Psi \rangle ,$$
we use the multiconfigurational time-dependent Hartree method for indistinguishable particles [19,20,21,38,50,71,72,73]. Regarding our notation in Equation (Equation 1), the MCTDH-X method implies that the field operators are represented by a sum of *M* time-dependent single-particle states or orbitals:(7)$$\hat{\Psi }\left(x\right)=\sum_{j=1}^{M}{\hat{b}}_{j}\phi_{j}\left(x;t\right).$$This corresponds to the following ansatz for the wavefunction:(8)$$|\Psi \rangle =\sum_{\vec{n}}C_{\vec{n}}|\vec{n};t\rangle .$$For bosons, the summation runs on all
N+M−1N
symmetric time-dependent configurations $\vec{n}=\left(n_{1},\cdots ,n_{M}\right)$ with fixed particle number $N=\sum_{i=1}^{M}n_{i}$. To derive the MCTDH-X equations, the ansatz in Equation (Equation 8) is plugged in a suitable variational principle [31,77,78,79,80,81]. The resulting equations are a two coupled sets—A set of linear equations for the coefficients $\left\{C_{\vec{n}}\left(t\right)\right\}$ and a set of non-linear ones for the orbitals $\left\{\phi_{j}\left(x;t\right);j=1,\cdots ,M\right\}$, see Appendix A for the equations and References [19,20,21,38,71,72,73] for details and derivation.

## 3. Quantities of Interest

Here, we define the observables that we use to quantify the dynamics of *N*-boson systems: the one-body density, the eigenvalues of the reduced one-body density matrix (1BDM), the angular momentum, and the image entropy and its variance evaluated from simulated single-shot images.
**Density, one-body density matrix, and natural occupations:**

The 1BDM is a hermitian matrix defined as
(9)$$\rho^{\left(1\right)}\left(x,x^{\prime }\right)=\langle \Psi |\hat{\Psi }\left(x\right){\hat{\Psi }}^{†}\left(x^{\prime }\right)|\Psi \rangle =\sum_{k,q}\rho_{kq}\phi_{k}^{*}\left(x;t\right)\phi_{q}\left(x;t\right).$$Here, we used the matrix elements $\rho_{kq}=\langle \Psi |{\hat{b}}_{k}^{†}{\hat{b}}_{q}|\Psi \rangle$ to represent the 1BDM using the orbitals corresponding to the creation and annihilation operators ${\hat{b}}_{k}^{†}$ and ${\hat{b}}_{q}$, respectively. The diagonal of the one-body density matrix is referred to as the density ρ(x):(10)$$\rho \left(x\right)=\rho^{\left(1\right)}\left(x,x^{\prime }=x\right).$$The density ρ(x) is a real quantity and has no phase, because it is the diagonal of a hermitian matrix, $\rho^{\left(1\right)}\left(x,x^{\prime }\right)$. The eigenvalues of the 1BDM, Equation (Equation 9), can be obtained via a diagonalization that corresponds to a unitary transformation of the orbitals $\phi_{j}\left(x;t\right)$ to the so-called natural orbitals $\phi_{j}^{\left(NO\right)}\left(x;t\right)$:(11)$$\frac{\rho^{\left(1\right)}\left(x,x^{\prime }\right)}{N}=\sum_{j}\lambda_{j}\phi_{j}^{(NO),*}\left(x;t\right)\phi_{j}^{\left(NO\right)}\left(x^{\prime };t\right).$$The eigenvalues $\lambda_{j}$ are normalized, $\sum_{j=1}^{M}\lambda_{j}=1$ and, without loss of generality, we consider them to be sorted by size, $\lambda_{1}\geq \lambda_{2}\geq \cdots$, throughout this work. The eigenvalues $\lambda_{j}$ (natural occupations) determine the degree of condensation and fragmentation of the system. Bosons with a 1BDM with only a single contributing eigenvalue $\lambda_{1}$ are condensed [82] and bosons with a 1BDM with multiple macroscopic eigenvalues contributing $\lambda_{1}∼O\left(N\right);\lambda_{2}∼O\left(N\right);\ldots$ are fragmented [60,61].

The eigenvalues of the 1BDM can be used as a precursor of correlations in the state |Ψ〉. This can be seen using, for instance, the Glauber first-order correlation function [83],
(12)$$|g^{\left(1\right)}\left(x,x^{\prime }\right)|=\left|\frac{\rho^{\left(1\right)}\left(x,x^{\prime }\right)}{\sqrt{\rho \left(x\right)\rho \left(x^{\prime }\right)}}\right|.$$If the system is in a condensed state, the 1BDM has only a single eigenvalue and is therefore a product of a single (complex-valued) orbital, $\rho^{\left(1\right)}\left(x,x^{\prime }\right)\propto \phi_{1}^{*}\left(x\right)\phi_{1}\left(x^{\prime }\right)$. It follows that $|g^{\left(1\right)}\left(x,x^{\prime }\right)|$ is constant for all $\left(x,x^{\prime }\right)$—for particles in a condensed state correlations are absent. It is instrumental to note here, that this condensed, single-orbital case with absent correlations is presupposed in mean-field approaches like the time-dependent Gross-Pitaevskii theory [17,18]. Similarly, when $\rho^{\left(1\right)}\left(x,x^{\prime }\right)$ is a sum of two or more orbitals as in Equation (Equation 9) then it can no longer be represented simply using its diagonal, the density $\rho^{\left(1\right)}\left(x,x^{\prime }=x\right)=\rho \left(x\right)$. Furthermore, in this fragmented case, the denominator of Equation (Equation 12) is a product of two weighted sums of orbitals, ρ(x), and $\rho (x^{\prime })$, respectively. This product involves non-trivial cross terms and the correlation function attains a value $|g^{\left(1\right)}\left(x,x^{\prime }\right)|\leq 1$ for all $\left(x,x^{\prime }\right)$—for particles in a fragmented state, correlations are present and the single-orbital picture of mean-field approaches like the time-dependent Gross-Pitaevskii theory [17,18] cannot be applied.
**Angular momentum:**

The angular momentum operator in ${\hat{e}}_{z}$-direction for a two-dimensional system is defined as
(13)$${\hat{L}}_{z}={\hat{e}}_{z}\cdot \left(\hat{x}\times \hat{p}\right)=-i\left(\hat{x}{\hat{\partial }}_{y}-\hat{y}{\hat{\partial }}_{x}\right).$$Bosonic quantum systems with angular momentum are rich in physics: they feature condensed vortices [17,18,84], phantom vortices [69,85], spatially partitioned many-body vortices [68,86], and fragmentation [68,69,85,86,87,88]. Since phantom vortices are the most pronounced characteristic feature of angular momentum which we find in our study below, we discuss them in the following. Typically, the term “vortex” refers to a topological defect in the density of quantum system connected with a discontinuity in the phase. A phantom vortex is an analog of such a conventional vortex, but for a natural orbital. A phantom vortex thus represents a topological defect connected with a discontinuity in the phase in one of the field modes of a many-particle states that corresponds to a natural orbital. Phantom vortices were shown to emerge as topological defects in the correlation function in Reference [69]. Moreover, in the common detection scheme for cold atoms, single-shot images (see below) they show as topological defects whose position fluctuates from image to image, see Reference [85].
**Single shots, image entropy and its variance:**

To assess the observability of the emergent physics in experimental setups with ultracold atoms, we simulate the detection of our numerical model wavefunctions in absorption or single-shot images [9,85,89]. A set of $N_{s}$ single shots,
(14)$$S^{j}=\left(s_{1}^{j},\ldots ,s_{N}^{j}\right);\;j=1,\ldots ,N_{s},$$
is nothing but $N_{s}$ random samples that are *N*-variate and distributed according to the *N*-particle probability given by ${\left|\Psi \right|}^{2}$,
(15)$$P\left(x_{1},\ldots ,x_{N}\right)={\left|\Psi \left(x_{1},\ldots ,x_{N}\right)\right|}^{2}.$$To generate images from these single shots, we convolute them with a point spread function. Typical choices include Gaussian (see [9,44,49,85]) or even quantum point spread functions [90]. Here, for simplicity, we consider the idealized case of a δ-shaped point spread function to obtain our single-shot images:(16)$$S^{j}\left(x\right)=\sum_{i=1}^{N}\delta \left(x-x_{i}^{j}\right).$$We will consider the image entropy ζ of single-shot images of the state |Ψ〉:(17)$$\zeta =-\frac{1}{N_{s}}\sum_{j=1}^{N_{s}}\zeta^{j};\;\zeta^{j}=\int dxS^{j}\left(x\right)\ln S^{j}\left(x\right).$$In the limit of large $N_{s}$, the image entropy ζ is equivalent to the density-entropy studied, for instance, in Reference [91]. Fundamentally, the image entropy is a measure for the information content in the particle distributions detected in single-shot images. While Ref. [91] found the entropy to be connected to the presence of correlations in the state, we found it not to be a conclusive pointer in our present work. The variances of observables, however, serve as a precursor of quantum fluctuations and correlations in many-body systems [39,40,41,42,43]; we are thus motivated to also analyze the variance $\sigma_{\zeta }$ of the image entropy ζ:(18)$$\sigma_{\zeta }=\frac{1}{N_{s}}\sum_{j=1}^{N_{s}}{\left[\zeta -\zeta^{j}\right]}^{2}.$$

## 4. Results

We now carve out the connection between artificial gauge fields and many-body correlations. For this purpose we focus on the dynamics of a model system of N=100 two-dimensional ultracold bosonic atoms with an interaction strength of $g_{0}=0.05$ [cf. Equations (Equation 1)–(Equation 5) for t≤0]. We now make an example for realizing this interaction $g_{0}$ with a real trap configuration with ${}^{87}$Rb in analogy to the study [69]—the Hamiltonian in Equation (Equation 1) is multiplied by $\frac{\hbar }{mL^{2}}$, where $m=1.44\times 10^{-25}$ kg is the mass of a ${}^{87}$Rb and *L* is a length scale that we choose to be $L=0.75\times 10^{-6}$ m. This sets our trapping frequency to ω=(2π)207 Hz and yields a unit of time, $\frac{mL^{2}}{\hbar }$ of 4.84 ms. The total interval of time we consider in the following t∈[0,200] thus corresponds to 0.97 s. In quasi-two-dimensional setups, the interaction parameter $g_{0}=2\sqrt{2\pi }\frac{a_{s}}{l_{z}}$ depends on the transversal oscillator length, $l_{z}=\sqrt{\frac{\hbar }{m\omega_{z}}}$, and the scattering length of ${}^{87}$Rb, $a_{s}=90.4a_{0}$; here, $\omega_{z}$ is the transversal trapping frequency and $a_{0}$ the Bohr radius. The two-dimensional interaction strength we use, $g_{0}=0.05$, is obtained for a transversal trapping of $\omega_{z}\approx \left(2\pi \right)3.178$ kHz. We remark here, that our choice of $g_{0}$ corresponds to a weak interaction; the healing length $\xi =\frac{1}{2g_{0}\left(N-1\right)}\approx 0.101$ is comparable to the oscillator length, that is, 1 in our units.

The system is initialized in its ground state and its dynamics (t>0) are then triggered by suddenly turning on an AMF of strength *B* [Equation (Equation 5)]. In what follows, we aim at an understanding of how the strength of the AMF affects the emergent dynamical behavior. For this purpose, in the main text, we solved the time-dependent many-body Schrödinger equation with MCTDH-X using M=4 orbitals [176,581 configurations in the state in Equation (Equation 8)] and 128×128 DVR functions (“modes”) to represent each of the orbitals $\left\{\phi_{j}\left(x;t\right)\right\}$; the total number of optimized and fully time-dependent parameters is thus $176581+4\times 128^{2}\times 4=242117$. For a convergence study with different *M* and *N* see Appendix B.

We open the exposition of our findings with the density ρ(x) and its decomposition into natural orbitals $\phi_{j}^{\left(NO\right)}$.

Besides a slight deformation of the density and the orbitals, little effects are seen in Figure 2a–e for a weak AMF, B=1.0. The phases $\beta_{1/2}\left(x\right)$, Figure 2f–g, hint there are no phantom vortices. The phases $\beta_{3/4}\left(x\right)$ feature topological defects aligned with zeros in $\phi_{3/4}\left(x\right)$, but these orbitals are occupied only by 0.1 particles.

For a comparatively strong AMF, B=6.5, in contrast, vortices at the edges of the density (so-called “ghost vortices” [92]) and phantom vortices [69] in the orbitals emerge in Figure 2j–r—zeros of the orbital densities are accompanied by topological defects in their phase [compare Figure 2, panels (k) and (o), (l) and (p), (m) and (q), (n) and (r), respectively].

These features of the density, orbitals, and their phases are hallmarks of the angular momentum that is deposited in time by the action of a sufficiently strong AMF—for increasing AMF strength *B*, the expectation value $L_{z}=\langle \Psi |{\hat{L}}_{z}|\Psi \rangle$ of the angular momentum operator ${\hat{L}}_{z}$ [Equation (Equation 13)] increases. For instance, we find $L_{z}/N=0.04$ at time t=50.0 and B=1.0 in Figure 2a–i and $L_{z}/N=1.26$ at time t=50.0 for B=6.5 in Figure 2j–r].

To quantify the dynamics of angular momentum triggered by quenches of the AMF a bit better, we plot $L_{z}/N$ for our system as a function of evolution time and as a function of the strength of the AMF in Figure 3.

We find from Figure 3a that a threshold AMF strength of about B≳6 is required to generate states with significant angular momentum at long evolution times (here, t=200). Furthermore, the average angular momentum content increases as the strength of the AMF does. We remark here that the angular momentum features an oscillatory behavior for all AMF strengths that we investigated. This can be understood as a consequence of our quench scenario—the initial state is an eigenstate of the Hamiltonian in Equation (Equation 1) for time t<0 and has vanishing angular momentum $L_{z}=0$. At t=0, the Hamiltonian and its spectrum is changed abruptly due to presence of the AMF, Equation (Equation 5). The dynamics of the initial state is thus dependent on these changes in the spectrum of the Hamiltonian by the AMF. At low AMF (B≲6) strength, the time-evolution of $|L_{z}|$ quasi-periodically goes back to $L_{z}=0$ and is dominated by oscillations at a single frequency [Figure 3b]; we infer that only very few states with $|L_{z}|\neq 0$ are contributing. The dynamics of the state |Ψ〉 of the many-boson system is thus a superposition of a very small number of eigenstates of the Hamiltonian after the quench in this quasi-adiabatic case.

The situation changes for B≳6, where the time-evolution of $|L_{z}|$ does *not* return to $L_{z}=0$ quasi-periodically. In this case, the time-evolution of $|L_{z}|$ still has its shortest-time oscillations of roughly the same amplitude as for the quasi-adiabatic case B≲6 [Figure 3c]. However, several large-amplitude oscillations with other frequencies contribute. We infer, that the dynamics of the state |Ψ〉 of the many-boson system is thus a superposition of a large number of states of the Hamiltonian after the quench, in this genuinely non-adiabatic evolution for B≳6.

We remark here that our results on the oscillatory behavior of the angular momentum in time render it impossible to approach the physics of the many-body state using a co-rotating frame at a certain angular frequency as done in References [62,93].

In References [62,68,69,85,87,88,93,94,95,96,97], an intricate connection of angular momentum content and the presence of correlations or the fragmentation of many-boson states has been pointed out. This motivates us to analyze the time-evolution of the eigenvalues of the reduced one-body density matrix as a precursor of correlations and the departure of the analyzed state from a mean-field description; we, thus, underpin the limitations of a mean-field description, see Figure 4 for a plot of $\lambda_{j}$ as a function of time and strength *B* of the AMF.

Our present findings for a quenched AMF are in line with results obtained with a time-dependent and slow transfer of angular momentum via a rotating asymmetry of the harmonic trapping potential [69,85]: the dynamical departure from a single-eigenvalue 1BDM to a fragmented many-body state with correlations accompanies the dynamical acquisition of a significant amount angular momentum. In particular, the region in time and AMF strength, with a quasi-adiabatic evolution of small $|L_{z}|$ in Figure 3 agrees roughly with the regions with small fragmentation in Figure 4. Conversely, the region in time and AMF strength, with a non-adiabatic evolution of large $|L_{z}|$ in Figure 3 agrees roughly with the regions with substantial fragmentation in Figure 4.

We now turn to the question of the possibility of an experimental detection of the emergent behavior of angular momentum and the eigenvalues of the one-body density matrix. For this purpose we simulated $N_{s}=1000$ single-shot images for all the many-body wavefunctions |Ψ(t)〉 for every time in t≡k·dt∈[0,200] in steps of dt=1.0. From this dataset of single-shot images, we computed the image entropy and its variance $\sigma_{\zeta }$ [Equation (Equation 18)]; see Figure 5.

The variance $\sigma_{\zeta }$ becomes significant for the same values of time and AMF strength where the natural occupations in Figure 4 herald the emergence of fragmentation and correlations. We emphasize here, that fragmentation implies the inapplicability of mean-field descriptions that are restricted to fully condensed and uncorrelated states, as already described in Section 3. By itself it is an interesting result that the variance of image entropy represents an experimentally feasible way to study the emergence of fragmentation in ultracold bosonic systems. Moreover, we observe that the region in time and AMF strength *B*, where $|L_{z}|$ features quasi-adiabatic (non-adiabatic) dynamics in Figure 3, coincides roughly with the region where fragmentation and image entropy are low (large) in Figure 4 and Figure 5, respectively. We therefore conjecture that an experimental detection of the presence of a non-adiabatic evolution of angular momentum is feasible by measuring the variance of the image entropy.

We remark that we have not shown complementary results on the image entropy ζ [Equation (Equation 17)] as a function of AMF strength and time, because it shows only very little overall variation. We noted that the entropy ζ is lowest, where its variance (Figure 5) is changing the most. The exploration of a possible fundamental connection of the gradient of the image entropy and the variance $\sigma_{\zeta }$ goes beyond the exploratory scope of our present investigation.

## 5. Conclusions and Outlook

We analyzed the dynamics of interacting two-dimensional ultracold bosonic particles triggered by a quench of an artificial gauge field. We used the multiconfigurational time-dependent Hartree method for indistinguishable particles software (http://ultracold.org) (accessed on π-day, 14 March 2021) to solve the many-body Schrödinger equation from first principles. Our exploratory investigation demonstrates that fragmentation emerges due to the quench if the artificial magnetic field is sufficiently strong. Such an emergence of fragmentation entails the breakdown of conventionally-used mean-field descriptions and, therewith, the occurrence of many-body correlations. We underpin our results by checking their consistency across different quality levels of our MCTDH-X approximation and different numbers of particles.

We have portrayed how correlations show in the expectation value of the angular momentum operator and in the orbitals and their phases as phantom vortices. Using simulations of single-shot images, we demonstrate that the fragmentation and correlations can be detected via the variance of the entropy of the images.

Our work highlights the importance of deploying modern computational and theoretical many-body approaches like the MCTDH-X to systems with artificial gauge fields as well as the necessity to consider not only the wavefunctions of ultracold atoms themselves, but also their detection.

Our results complement the recent findings, that the variances of observables are sensitive probes of correlations in the state of ultracold atomic systems [9,38,39,40,41,42,43,96,98].

Straightforward continuations of this investigation include the deployment of the developed analysis and computational tools to other many-body systems. As examples of interest, we name here the variance of the image entropy in ultracold dipolar atoms as discussed in Refs. [44,48,49] and an exploration of the competition of long-ranged dipolar interactions and artificial magnetic fields for two-dimensional ultracold atoms. Another direction of physical interest is the dynamics of two-dimensional bosonic Josephson junctions [99] subject to gauge fields and the resulting tunneling of (phantom) vortices or the emergence of quantum turbulence [100] via entropy production [101], which, in turn, as we have shown above, could result from the presence of artificial gauge fields.

## Figures and Tables

**Figure 1 entropy-23-00392-f001:**
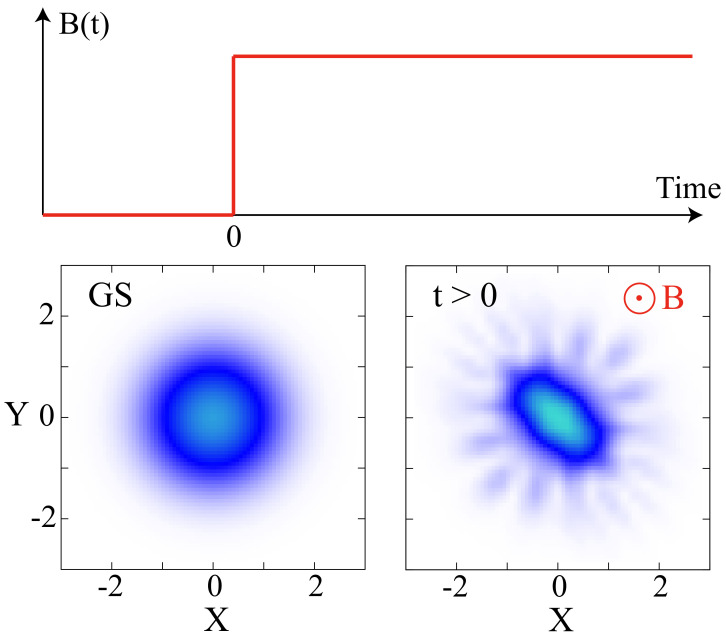
**Sketch of our setup.** Two-dimensional ultracold bosonic particles are prepared in the ground state of an isotropic harmonic trap (label “GS”) for time t≤0. At t>0, an artificial magnetic field pointing in the *Z*-direction perpendicular to the ultracold atoms of strength B(t) (top) is switched on suddenly. This quench triggers many-body dynamics of the state (label “t>0”).

**Figure 2 entropy-23-00392-f002:**
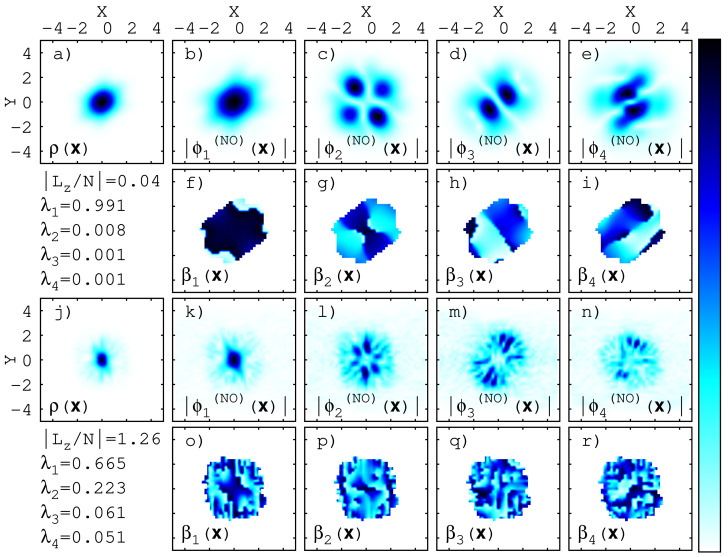
**One-body density, natural orbitals, and natural orbital phases** for two distinct AMF strengths: B=1.0 (weak) in (**a**–**i**) and B=6.5 (strong) in (**j**–**r**). The figure shows the quantities at time t=50.0. For guidance, the angular momentum per particle and the natural occupations are listed below (**a**) and (**j**) for weak and strong AMFs, respectively. The phase plots [$\beta_{j}\left(x\right)$ in (**f**–**i**) and (**o**–**r**)] are restricted to areas where ρ(x)>0.01. The plot ranges are [0,∼0.2] for (**a**) and (**j**), [0,∼0.1] for (**b**–**e**) and (**k**–**n**), and [−π,π] for (**o**–**r**). See text for further discussion.

**Figure 3 entropy-23-00392-f003:**
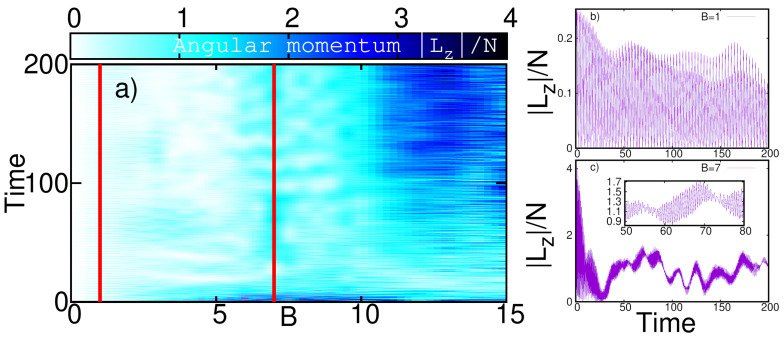
**Angular momentum as a function of propagation time and strength of artificial magnetic field (AMF).** The expectation value $L_{z}/N=\langle \Psi \left(t\right)|{\hat{L}}_{z}|\Psi \left(t\right)\rangle /N$ [cf. Equation (Equation 13)] is shown as a function of time *t* and AMF strength *B* in (**a**). Interestingly, there is a drastic short-term increase following a subsequent equilibration of $L_{z}/N$ for 5≲B≲12. To highlight the oscillatory nature of the angular momentum we plot the cuts at B=1 and B=7 [highlighted by the red vertical lines in a)] in panels (**b**,**c**), respectively. The angular momentum returns to $L_{z}=0$ quasi-periodically for B≲6 [as in (**b**)], but not for B≳6 [as in (**c**)]. See text for discussion.

**Figure 4 entropy-23-00392-f004:**
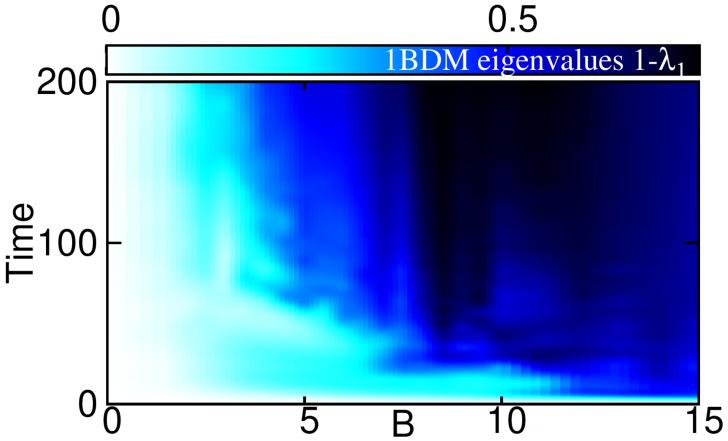
**Eigenvalues of the reduced one-body density matrix.** The dynamical emergence of multiple significant eigenvalues, i.e., the fragmentation of the many-body state, is quantified here via the time-dependent depletion $1-\lambda_{1}=\sum_{k=2}^{M}\lambda_{k}$ and is in sync with the dynamics of angular momentum (cf. Figure 3).

**Figure 5 entropy-23-00392-f005:**
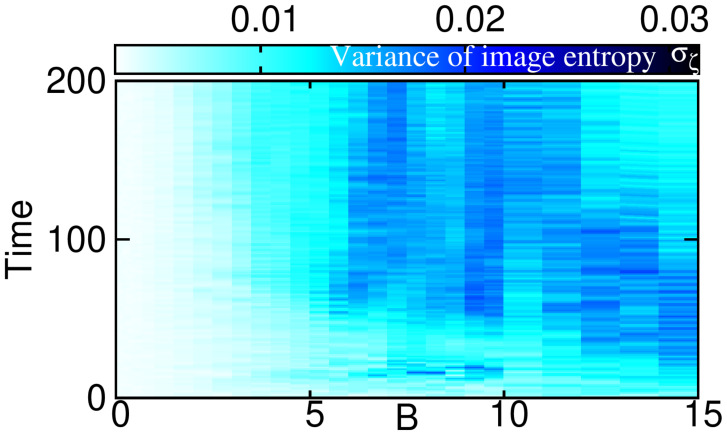
**Variance of the image entropy as a function of propagation time and strength of the AMF.** The variance of image entropy $\sigma_{\zeta }$ tracks the behavior of the natural occupations and the angular momentum in, respectively, Figure 3 and Figure 4 closely, see text for discussion.

## Data Availability

The MCTDH-X software to recompute all the results in this study is available at http://ultracold.org (accessed on π-day, 14 March 2021). Results can be made available upon request; the total amount of data is several TB.

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
