# Peer review of "Dynamics of Ultracold Bosons in Artificial Gauge Fields—Angular Momentum, Fragmentation, and the Variance of Entropy"

_entropy, 2021, doi:10.3390/e23040392_

Round 1

Reviewer 1 Report

The manuscript by Lode et al. describes their numerical investigation into the behaviour of a two-dimensional Bose gas at ultra-low temperatures after a magnetic field is quenched on.  The authors use a many-body approach, that allows them to get insight into the correlations that are being built up as a consequence of this quench.

While this is in principle in an interesting topic, the manuscript presented unfortunately suffers from a large number of shortcomings and is not up to scientific standards. Let me stress that I have no reason to doubt the correctness of the advanced numerical calculations, however the interpretation falls short in many aspects. Let me point out a few below.

  • at no point in the manuscript is the convergence of the numerical tool discussed. While for experts in the area of MCTBDH-B this might be an answered problem, for the more broader readership of Entropy it is worth pointing out how the finite size of the many-body basis can potentially effect the results.
  • the above point is also connected to the use of the point-like interaction. I understand that the incomplete basis leads to a scattering effect and therefore to a useable interaction, but the choice of \lambda_0=0.01 in natural units seems utterly random. The authors do not even discuss which regime they are in here, i.e. is it close to a non-interacting BEC or close to the TF regime? Given the small number of modes that can be numerically treated, I assume it is close to the non-interacting regime? Additionally, calling the interaction strength \lambda_0 is not the best choice when the 1BDM eigenvalues are called \lambda_1 etc.
  • while in principle one could learn something about the regime the work is in from the density distributions in Fig. 1, the choice to display rotationally symmetric GS as a 3D plot, unfortunately does not allow for this. In general the value of Fig. 1 is quite questionable, as the upper part just shows what a Heaviside function is, while the lower one just shows the GS (without one being able to tell anything about it, see comment above) on the left and some other state on the right, which the readers are left to assume is the GS in the presence of a B field?  It is clearly not the state indicated for  t>0 with the B field dynamics indicated in the graph above. This figure needs quite some work to be useful.
  • The discussion of the results shown in Fig. 2 is not clear. The authors point out that vortices can be seen in the higher order natural orbitals once the strength of the B-field exceeds a certain threshold, but it is not clear what the meaning of having a singularity in a natural orbital is? How does this translate into the vortices in the physically relevant system is the full density (shown in panels (a) and (j))? Moreover, the density scale is chosen so poorly that the vortices are hardly visible and no phase plot for the full density situation is given. At least choosing a better colour scale could help the reader see what is going on in panel (j). 
  • Fig. 3 shows the angular momentum as a function of time after the quench and the strength of the B field. This is an interesting quantity, however again this plot is hardly discussed in the text apart from noting two areas with strong angular momentum. The clearly visible oscillations and the transitory regions are not addressed at all. The only takeaway from this plot (according to the authors) is that applying a larger amount of B leads to more vortices. This is hardly enough to justify all the numerical work done.
  • The idea behind Fig. 4 is in principle interesting, however the authors do not really explore what is going on here, apart from pointing out a ‘similarity’ with Fig. 3. In fact, they claim that Figs. 3 and 4 are ‘in sync’ (which in itself is a rather unscientific term), which is not obvious to me. The strongly correlated area visible around T>150 for B between 9 and 12 has no real analogy in Fig. 3.
  • The authors also use Fig. 4 to claim that the system is strongly correlated after the quench. Again this is not clear to me, as the plot shows 1-\lambda_1, which tells nothing really about the values of the other eigenvalues. In fact, in the mean-field limit a condensate can be fully coherent and carry vortices. Similarly, one could consider it possible that in the region where 1-\lambda_1 goes to zero, simply \lambda_2 goes towards 1, which would not be an indication for strong correlations. This could be checked by going into a co-rotating frame (where the rotating state would become the ground state), but I don’t think the authors are doing this here. 
  • For the variance shown in Fig. 5 the authors again claim that this plot ‘tracks’ the results of Figs. 3 and 4 ‘closely’. While one could maybe (very strong maybe) argue this for Fig. 4, this is clearly not true for Fig. 3.
  • In the conclusions the authors claim that ‘many-body correlations emerge due to a quench if the artificial magnetic field is sufficiently strong’. I don’t think this is shown in their work at all. Or if their simulations show this, it is not worked out sufficiently.
  • I am not sure what to say about Fig. 6, apart from urging the authors to choose a colour scale that actually allows the readers to see something. If, as the authors state in the Appendix, the interpretation of the image entropy in terms of what is going on goes beyond the work they want to do, why show this plot at all? What are the readers supposed to learn here?

Finally, this us a very brief manuscript that presents a large amount of numerical calculations carried out. Unfortunately, it falls much too short when it comes to interpreting the results in physical terms. It was very surprising to me that such a short work would have about 100 citations and on a closer look a lot of them seem only very vaguely related to the current work. In particular, more than a quarter of the citations (25) are papers of work of the first author on generally related works, however by no means helpful for understanding the current work. See in particular the [39-55] batch of citations for exploring strong correlations using MCTDH. Or the several citations for using the code hosted at ultracold.org. Adding such a large number of self-citations, quite a few of which are arguably not directly related, suggests that the authors see this work more as an advertisement for their other works, and do not really value to time of potential readers or referees. One could argue even stronger here, but I will trust in the integrity of the authors that any future re-submission will address this issue as well.

Reviewer 2 Report

Report

In the manuscript, the authors have studied the dynamics of an ultracold bosonic system in artificial gauge field using the MCTDH_X method. The authors performed dynamical calculations for a quench in artificial gauge field at t=0 and evaluated the system properties such as the density, one-body density matrix, angular momentum and the image entropy. I find the article is well written and interesting to read.

However, I have the following comments which the authors should address:

  1. I would suggest the authors include a definition of what they mean by image entropy. It will be a good idea if the definition along with the physical significance of this quantity is explained.
  2. It will also be helpful if the authors address the relevance and importance of the work more clearly. I understand that their method takes into account the correlations neglected in the usual mean-field study, but a clear explanation of the advantages of their method and comparison of their results with mean-field would be better.

Once the authors address the above-mentioned comments, I can accept the article for publication.

Reviewer 3 Report

The authors have studied the dynamics of two-dimensional ultracold bosons. The authors state that the manipulation of some physical parameters such as the angular momentum can control the dynamics. 

The paper needs major revision, my comments are:

1) The authors wrote:

"Formally, one cannot use a contact interaction in two spatial dimensions with a complete basis set since the outcome would be that of the noninteracting bosons."

This should be clarified, an appendix can be added to it.

2) The authors stated that: "aim to demonstrate that beyond-mean-field
phenomena do emerge"  they need to give some physical examples where

some previous studies have shown emerging new phenomena beyond mean-field approximation to increase the visibility of the paper.

3) The MCTDH-X method should be presented. It could be discussed in an appendix.

4) There is no code color for figs 3,4,5 and 6.

5) The paper needs more discussions on the results

6) Several important details that clarifying for the readers the results are hidden.

Round 2

Reviewer 1 Report

Clearly the authors and I disagree about what constitutes a helpful physical interpretation and discussion of a numerical result. The response to my report has led to some effort by the authors to more carefully describe their numerical results, but very little has been done to carefully understand them in detail; very little has been done to improve the accessibility of the figures.

That fragmentation happens is well understood effect and for me understanding how it happens in this specific system would have been what adds value to the numerical simulations. The detailed out-of-equilibrium dynamics, to which the authors have access in their simulations, could have been investigated and explained in a lot more detail, with specific examples discussed in different regimes. I understand and appreciate the large and difficult numerical efforts that were involved in obtaining the data, but to me the value of its current presentation is low.

However, there is no reason to assume that my opinion on this is shared by all readers, and since I do not consider anything to be wrong in this work, I am not objecting publication.

Finally, even though the authors argue this point (not very convincingly), there is no doubt that this manuscript contains an excessive amount of self-citations. Apart from the usual worries about this, it is also another example of an aspect of the manuscript that has only received low effort.

I really hope that a future work will more carefully treat this very interesting quench dynamics.

Reviewer 2 Report

After carefully going through the revised version of the manuscript, I recommend the publication of this article in its present form.

Reviewer 3 Report

There is a need for the second round of revision where the authors should take into consideration all the previsous comments.